# CHIP Haploinsufficiency Exacerbates Hepatic Steatosis via Enhanced TXNIP Expression and Endoplasmic Reticulum Stress Responses

**DOI:** 10.3390/antiox12020458

**Published:** 2023-02-11

**Authors:** Jung-Hwa Han, Dae-Hwan Nam, Seon-Hui Kim, Ae-Rang Hwang, So-Young Park, Jae Hyang Lim, Chang-Hoon Woo

**Affiliations:** 1PNU GRAND Convergence Medical Science Education Research Center, Pusan National University School of Medicine, Yangsan 50612, Republic of Korea; 2Immune Research Institute, Seegene Medical Foundation, Seoul 04805, Republic of Korea; 3Department of Pharmacology, Yeungnam University College of Medicine, 170 Hyeonchung-ro, Nam-gu, Daegu 42415, Republic of Korea; 4Department of Physiology, Yeungnam University College of Medicine, 170 Hyeonchung-ro, Nam-gu, Daegu 42415, Republic of Korea; 5Senotherpy-Based Metabolic Disease Control Research Center, Yeungnam University College of Medicine, 170 Hyeonchung-ro, Nam-gu, Daegu 42415, Republic of Korea; 6Department of Microbiology and Ewha Education and Research Center for Infection, Ewha Womans University College of Medicine, 25 Magokdong-ro 2-gil, Seoul 07804, Republic of Korea

**Keywords:** carboxyl terminus of the Hsc70-interacting protein (CHIP), thioredoxin-interacting protein (TXNIP), non-alcoholic fatty liver disease (NAFLD), endoplasmic reticulum (ER) stress, metabolic disease

## Abstract

TXNIP is a critical regulator of glucose homeostasis, fatty acid synthesis, and cholesterol accumulation in the liver, and it has been reported that metabolic diseases, such as obesity, atherosclerosis, hyperlipidemia, type 2 diabetes, and nonalcoholic fatty liver disease (NAFLD), are associated with endoplasmic reticulum (ER) stress. Because CHIP, an E3 ligase, was known to be involved in regulating tissue injury and inflammation in liver, its role in regulating ER stress-induced NAFLD was investigated in two experimental NAFLD models, a tunicamycin (TM)-induced and other diet-induced NAFLD mice models. In the TM-induced NAFLD model, intraperitoneal injection of TM induced liver steatosis in both CHIP^+/+^ and CHIP^+/−^ mice, but it was severely exacerbated in CHIP^+/−^ mice compared to CHIP^+/+^ mice. Key regulators of ER stress and de novo lipogenesis were also enhanced in the livers of TM-inoculated CHIP^+/−^ mice. Furthermore, in the diet-induced NAFLD models, CHIP^+/−^ mice developed severely impaired glucose tolerance, insulin resistance and hepatic steatosis compared to CHIP^+/+^ mice. Interestingly, CHIP promoted ubiquitin-dependent degradation of TXNIP in vitro, and inhibition of TXNIP was further found to alleviate the inflammation and ER stress responses increased by CHIP inhibition. In addition, the expression of TXNIP was increased in mice deficient in CHIP in the TM- and diet-induced models. These findings suggest that CHIP modulates ER stress and inflammatory responses by inhibiting TXNIP, and that CHIP protects against TM- or HF–HS diet-induced NAFLD and serves as a potential therapeutic means for treating liver diseases.

## 1. Introduction

Non-alcoholic fatty liver disease (NAFLD) includes a variety of pathologies ranging from hepatic steatosis to non-alcoholic steatohepatitis (NASH), which can progress to liver fibrosis, cirrhosis, or hepatocellular carcinoma [1,2]. NAFLD is characterized by excessive triglyceride accumulation in the liver resulting in type 2 diabetes, hyperlipidemia, obesity, or insulin resistance [3,4]. Multiple physiological and pathological pathways, including endoplasmic reticulum (ER) stress, oxidative stress, iron accumulation, and endotoxins and cytokines, contribute to the development of inflammation, cellular injury, and fibrosis in the pathogenesis of NAFLD [5,6]. However, the molecular mechanisms responsible for NAFLD progression are not fully understood yet.

The ER is the largest cellular organelle and performs various functions including protein folding, calcium storage, biosynthesis of macromolecules, lipid and steroid synthesis, and removal of toxins [7]. ER stress can be induced by pharmacological and pathophysiological stimuli, which can increase the accumulation of unfolded proteins in the ER [8,9]. Unfolded and misfolded proteins bind to and sequester the ER chaperone BiP (also known as 78 kDa glucose-regulated protein, GRP78) thereby activating the unfolded protein response (UPR) to alleviate ER stress and restore ER homeostasis [10]. Inositol-requiring protein 1 (IRE1)/X-box binding protein 1 (XBP1), PRKR-like endoplasmic reticulum kinase (PERK)/eukaryotic translation initiation factor 2a (eIF2a), and activating transcription factor-6 (ATF6) are three main UPR-sensing proteins, playing critical roles in ER stress responses [11]. Metabolic diseases, such as atherosclerosis, obesity, dyslipidemia, type 2 diabetes, and NAFLD, are reported to be associated with ER stress [12,13,14]. Tunicamycin (TM) is a bacterial nucleoside antibiotic, which can block N-linked glycoproteins thereby resulting in the accumulation of misfolded and unfolded proteins in ER [15,16], and thus, is commonly used to induce ER stress. In fact, several studies have demonstrated that intraperitoneal injection of TM in mice efficiently induces ER stress in liver and kidney [13,16,17]. Furthermore, it has been suggested that ER stress exacerbates NASH by regulating hepatic lipid metabolism and inflammation [16].

Thioredoxin-interacting protein (TXNIP, also known as vitamin D3 up-regulated protein-1, VDUP-1, or thioredoxin binding protein 2, TBP-2) directly binds to the two active cysteine residues of thioredoxin, interacts with its active catalytic site, and suppresses its expression and activity [18,19]. TXNIP is also involved in a variety of cellular processes, including cell proliferation, oxidative stress, and apoptosis by negatively regulating thioredoxin activity [20]. Recent studies have shown that TXNIP is a critical signal that links inflammation and ER stress to cancers, cardiac diseases, and other metabolic diseases [20,21,22]. TXNIP is also a critical regulator of glucose homeostasis, cholesterol accumulation, and fatty acid synthesis in the liver [19]. Thus, TXNIP has emerged as an important hepatic factor, and its regulation is believed to be necessary for the maintenance of liver function.

It has been suggested that NAFLD is linked to hepatic manifestations of metabolic syndrome such as type 2 diabetes, hyperlipidemia, hypertension, and obesity [23]. In particular, hepatic steatosis is primarily a result of excessive caloric intake, lack of physical activity, and an unhealthy lifestyle which should be corrected as a first step to the prevention and treatment of NAFLD [23]. In terms of molecular mechanisms of high-fat diet (HFD)-induced obesity-associated NAFLD, mitochondrial dysfunction has been associated with metabolic abnormalities in the development of hepatic steatosis [24]. In addition, supplementation with docosahexaenoic acid hydroxytyrosol improved mitochondrial respiratory function and progression into NASH, suggesting that improvement of mitochondrial dysfunction might be an important strategy for the treatment of NAFLD.

Ubiquitination is a type of post-translational modification, in which ubiquitin bonds to a substrate protein. Ubiquitination of target substrates signals proteins for degradation via the proteasome, alters cellular locations, changes their activities, and regulates protein–protein interactions. The ubiquitination process involves three steps, namely, activation, conjugation, and ligation, which are performed by E1 ubiquitin-activating enzymes, E2 ubiquitin-conjugating enzymes, and E3 ubiquitin ligases, respectively [25]. Carboxyl terminus of the Hsc70-interacting protein (CHIP) is an E3 ubiquitin ligase that possesses a carboxyl terminus with U-box-dependent ubiquitin ligase activity and cochaperone/chaperone activity [26,27]. CHIP plays an important role in the control of protein quality by integrating with the molecular chaperone machinery and the ubiquitin–proteasome system to regulate protein folding homeostasis under proteotoxic stress [28]. Previous studies have demonstrated that CHIP is an E3 ubiquitin ligase for ER-associated degradation (ERAD), a process involved in cellular adaptations to ER stress [29,30]. In addition, CHIP has been reported to prevent ER stress-induced cell death in the central nervous system [31]. Furthermore, accumulating evidence indicates that CHIP is involved in metabolic pathways including cardiac dysfunction, lung inflammation, liver injury, and NASH [32,33,34,35]. It has also been reported that CHIP^−/−^ mice have shorter lives and exhibit an accelerated aging phenotype, partial perinatal lethality, and altered protein quality control [36,37]. These findings suggest that CHIP regulates ER stress-associated metabolic disorders. Thus, we hypothesized that CHIP might regulate TXNIP expression and might be involved in the development of TM- or diet-induced NAFLD in mice. Specifically, we aimed to elucidate the role of CHIP in the regulation of TXNIP and ER stress-mediated NAFLD in mice.

## 2. Materials and Methods

### 2.1. Cell Culture

The human hepatoma cell line HepG2 (ATCC) was cultured in Dulbecco’s Modified Eagle Medium (DMEM, Welgene Inc., Daegu, Republic of Korea) supplemented with 10% fetal bovine serum (FBS, Welgene Inc. Kyungsan, Republic of Korea), 50 U/mL penicillin, and 50 μg/mL streptomycin (Welgene Inc. Kyungsan, Republic of Korea). The mouse hepatocyte cell line AML12 (ATCC) was cultured in DMEM/F-12 medium supplemented with 10% bovine calf serum (Welgene Inc. Kyungsan, Republic of Korea), 1× insulin–transferrin–selenium–pyruvate supplement (ITSP, Welgene Inc. Kyungsan, Republic of Korea), 1 nM dexamethasone (Sigma, St. Louis, MO, USA), 100 nM insulin (Sigma Aldrich, St. Louis, MO, USA), 50 U/mL penicillin, and 50 μg/mL streptomycin (Welgene Inc. Kyungsan, Republic of Korea). Primary hepatocytes were isolated from 6–8-week-old male mice as previously described [38]. Briefly, the liver was flushed with perfusion buffer (HBSS with EDTA and HEPES) through the portal vein, dissociated with collagenase, and excised and ruptured with fine tip forceps. The hepatocytes were gently released, filtered through a 70 μm cell strainer into a 50 mL tube, and collected by centrifugation at 50× *g* for 2 min at 4 °C. The hepatocytes were seeded on collagen-coated plates in DMEM supplemented with 10% FBS, 15 mM HEPES (Welgene Inc. Kyungsan, Republic of Korea), 100 nM dexamethasone, 50 U/mL penicillin, and 50 μg/mL streptomycin. The cells were incubated in a humidified atmosphere containing 5% CO_2_ at 37 °C.

### 2.2. Animal Experiments

All the animal experiments were approved by the Institutional Animal Care and Use Committee of Yeungnam University College of Medicine (Daegu, Republic of Korea). CHIP^+/−^ mice, generated as described previously [36], were generously provided by Prof. Cam Patterson (University of North Carolina, Chapel Hill, North Carolina). For the tunicamycin (TM, Sigma Aldrich)-induced NAFLD model, 8-week-old female CHIP^+/+^, CHIP^+/−^, and CHIP^−/−^ mice (129/SvEv × C57BL/6 background) were intraperitoneally (i.p.) injected with TM (2 μg/g body weight). Blood samples and liver tissues were collected 8 h or 36 h after TM injection.

For the diet-induced hepatic steatosis model, two different types of diets were used. For the high-fat–high-sucrose (HF–HS) diet-induced NAFLD model, 8-week-old male CHIP^+/+^ or CHIP^+/−^ mice were fed either company-recommended standard chow (11.5% fat, 0% sucrose, DooYeol Biotech, Seoul, Republic of Korea) or HF–HS diet (36% fat, 30% sucrose, DooYeol Biotech) for 22 weeks. For the high-fat (HF) diet-induced NAFLD model, the mice were fed company-recommended standard chow (10% fat, Research Diets, New Brunswick, NJ) or HF diet (60% fat, Research Diets) for 12 weeks. Body weight was measured weekly, and livers and fat pads were collected at the end of experiments. Fat mass was determined by measuring gonadal fat pad weight.

### 2.3. Quantitative Real Time RT-PCR

mRNA levels were determined by quantitative real time RT-PCR (qRT-PCR). Briefly, total RNA was extracted using TRIzol reagent (Invitrogen, Carlsbad, CA, USA), and reverse transcription reaction was conducted using TaqMan reverse transcription reagents (Applied Biosystems, Carlsbad, CA, USA), according to the manufacturer’s instructions. qRT-PCR was conducted with 1 μg of template cDNA and Power SYBR Green PCR Master Mix (Applied Biosystems) using an ABI PRISM 7500 (Applied Biosystems). Quantification was performed using the efficiency-corrected ΔΔCt method. The primers used to amplify DNA sequences were as follows: mouse ACC (NM_133360.2), forward 5′-GCGGGAGGAGTTCCTAATTC-3′ and reverse 5′-TGTCCCAGACGTAAGCCTTC-3′; mouse ATF4 (NM_009716.3), forward 5′-TGGAAACCATGCCAGATGAG-3′ and reverse 5′-GATGGCCAATTGGGTTCACT-3′; mouse ATF6 (NM_001081304.1), forward 5′-TCGCCTTTTAGTCCGGTTCTT-3′ and reverse 5′-GGCTCCATAGGTCTGACTCC-3′; mouse APOB (NM_009693.2), forward 5′-CAGCCATGGGCAACTTTAC-3′ and reverse 5′-TGGGCAACGATATCTGATTG-3′; mouse CD36 (NM_001159558.1), forward 5′-GAGGAGAATGGGCTGTGATC-3´ and reverse 5′-GTCTCCGACTGGCATGAGA-3′; mouse CHOP (NM_007837.4), forward 5′-GCATGAAGGAGAAGGAGCAG-3′ and reverse 5′-CTTCCGGAGAGACAGACAGG-3′; mouse FATP1 (NM_011977.4), forward 5′-GGCGTTTCGATGGTTATGT-3′ and reverse 5′-AGCACGTCACCTGAGAGGTA-3′; mouse IL-6 (NM_031168.2), forward 5′-GCTACCAAACTGGATATAATCAGGA-3′ and reverse 5′-CCAGGTAGCTATGGTACTCCAGAA-3′; mouse IL-1β (NM_008361.4), forward 5′-TTGACGGACCCCAAAAGAT-3′ and reverse 5′- GATGATCTGAGTGTGAGGGTCTG-3′; mouse PPARα (NM_011144.6), forward 5′-AGGCTGTAAGGGCTTCTTTC-3′ and reverse 5′-GCATTTGTTCCGGTTCTTCT-3′; mouse TNFα (NM_013693.3), forward 5′-CTACTCCCAGGTTCTCTTCAA-3′ and reverse 5′-GCAGAGAGGAGGTTGACTTTC-3′; and mouse GAPDH (NM_001289726) forward 5′-GGAGCCAAAAGGGTCATCAT-3′ and reverse 5′-GTGATGGCATGGACTGTGGT-3′.

### 2.4. Immunoprecipitation and Immunoblotting

The cells were lysed with radioimmunoprecipitation assay (RIPA) lysis buffer supplemented with 1 mM phenylmethylsulfonyl fluoride (PMSF) and 0.01 mM protease inhibitor cocktail (PIC) on ice for 15 min and centrifuged at 15,000× *g* for 10 min at 4 °C. The protein concentration was determined using the Bradford assay. For immunoprecipitation, cell lysates were incubated with rabbit anti-CHIP antibody overnight at 4 °C followed by 1 h incubation with protein A–agarose beads (Invitrogen) on a roller system at 4 °C. The beads were collected by centrifugation and then washed with washing buffer (50 mM Tris-HCl, pH 7.4, 0.1% Nonidet P-40, 150 mM NaCl, and 1 mM EDTA). Bound proteins were released using 2× sodium dodecyl sulfate (SDS) sample buffer. Proteins were separated by SDS-polyacrylamide gel electrophoresis (SDS-PAGE) and transferred to polyvinylidene difluoride membranes. The membranes were blocked with 5% skimmed milk and immunoblotted with primary antibodies overnight at 4 °C and then with corresponding secondary antibodies at room temperature (RT) for 1 h. The protein signals were visualized by using electrochemiluminescence detection reagents (Millipore, Billerica, MA, USA) according to the manufacturer’s instructions. The antibodies were purchased from the following vendors: TXNIP (MBL International, Woburn, MA, USA); KDEL (GRP94, GRP78) (Enzo Life Sciences, Lörrach, Germany); XBP-1s (BioLegend, San Diego, CA, USA); ATF6 (Novus Biologicals, Littleton, CO, USA); ATF4, GADD153 (CHOP), HA, and CHIP (Santa Cruz Biotechnology, Santa Cruz, CA, USA); Akt, p-Akt, ACC, FAS, PARP-1, and cleaved Caspase-3 (Cell Signaling, Danvers, MA, USA); NLRP3 (ThermoFisher, Waltham, MA, USA); PGC1α (abcam, Cambridge, UK); and α-tubulin and β-actin (Sigma Aldrich, St. Louis, MO, USA).

### 2.5. RNA Interference

For CHIP or TXNIP silencing, cells were transiently transfected with control small interfering (siRNA) or siRNA against CHIP or TXNIP using Lipofectamine 2000 (Invitrogen) according to the manufacturer’s instructions. Mouse CHIP siRNA and non-specific control siRNA were purchased from Bioneer (Daejeon, Republic of Korea). TXNIP siRNAs (sc-44944 for mouse, sc-44943 for human) were purchased from Santa Cruz Biotechnology (Santa Cruz, CA). The mouse and rat specific CHIP target sequence was 5´-GGGAUGAUAUUCCUAGUGC-3´. Non-specific control siRNA was used as the negative control. The cells were harvested 48 to 72 h after siRNA transfection, and protein expression was determined by immunoblotting with specific antibodies.

### 2.6. Histological Analysis

Liver tissues were fixed in 10% buffered formalin, paraffin-embedded, and cut into 5 μm slices. The sections were subjected to hematoxylin and eosin (H&E) staining and immunohistochemistry for TXNIP expression. The sections were deparaffinized, rehydrated, blocked, and then incubated with anti-TXNIP antibody (MBL International). The immunohistochemistry was performed using the HRP/DAP (ABC) detection IHC kit (Abcam, Cambridge, UK) following the manufacturer’s instructions. For Oil Red O staining, liver tissues were embedded in optimum cutting temperature compound (OCT, Sakura Finetek, Zoeterwoude, Netherlands), snap-frozen, and then cryosectioned. Frozen liver sections were stained with Oil Red O solution (Sigma Aldrich) for 30 min, rinsed in 60% isopropanol, and then washed with water. All the microscopic images of sections were obtained using an optical microscope (Nikon, Tokyo, Japan).

### 2.7. Intraperitoneal Glucose Tolerance Test, Intraperitoneal Insulin Tolerance Test, and Insulin Sensitivity Assessment

Intraperitoneal glucose tolerance tests (IPGTTs) and intraperitoneal insulin tolerance tests (IPITTs) were performed at the end of the diet feeding schedule (HF–HS diet for 22 weeks, HF diet for 12 weeks). For IPGTT, the mice were fasted for 6 h with free access to water, and then glucose (2 g/kg body weight i.p.) was administered. For IPITT, the mice were fasted for 4 h with free access to water, and then insulin (0.3 U/kg body weight i.p.) was administered. Blood samples were collected by tail tip puncture at 0, 15, 30, 60, 90, or 120 min after glucose or insulin injection for glucose analyses using a glucometer (Accu-Chek; Roche, Indianapolis, IN). To examine insulin sensitivity, HF diet-fed mice were administered insulin (1.5 U/kg body weight i.p.) and sacrificed 10 min after insulin injection. Liver tissues were immediately harvested, homogenized, lysed, and immunoblotted using phospho-Akt and Akt antibodies.

### 2.8. Oxidative Status Analysis

Oxidation stress markers were analyzed in liver tissue using the ELISA method. Liver tissues were homogenized in cooled PBS/cell lysates, and then the supernatant was taken as the test sample after centrifugation. By using the assay kits (Cayman Chemical, Ann Arbor, MI, USA), we detected the levels of malondialdehyde (MDA) in the tissues. The homogenized liver tissue was centrifuged at 1600× *g* for 10 min at 4 °C, then the supernatant was used for biochemical analysis. The color intensity was measured at 530~540 nm. All steps were carried out strictly following the kit instructions.

### 2.9. Ubiquitination Assay

HepG2 cells were transfected with HA-tagged ubiquitin (Addgene #11928), GFP-tagged TXNIP (Addgene #18758), Myc-tagged CHIP WT, or Myc-tagged CHIP mutant (H260Q) and incubated for 24 h (CHIP, GeneBank AF 129085.1). The cells were lysed in RIPA buffer containing 5 mM NEM (N-ethylmaleimide), 1 mM PMSF, and 0.01 mM PIC, and centrifuged at 13,000× *g* for 10 min. The cell lysates were incubated with mouse anti-TXNIP antibody overnight at 4 °C and then incubated with protein A–agarose beads for 1 h on a roller system at 4 °C. The beads were collected by centrifugation and washed with washing buffer. The bound proteins were released using 2× SDS sample buffer. The immunoprecipitates were separated by SDS-PAGE, and the levels of ubiquitinated forms of TXNIP were assessed by immunoblotting with anti-HA antibody.

### 2.10. Statistical Analysis

The results in the bar graphs were expressed as the mean ± S.D. of three independent experiments. The statistical analysis was performed using Student’s t test or ANOVA followed by Bonferroni post hoc tests for multiple group comparisons using GraphPad prism 8.0 (Graph-Pad Software Inc., Boston, MA, USA). Probability values (*p* values) of <0.05 were considered statistically significant.

## 3. Results

### 3.1. CHIP Is Involved in Unfolded Protein Responses and Apoptosis In Vitro

To determine whether CHIP mediates ER stress and apoptosis in vitro, AML12 cells were transfected with control siRNA (siControl) or CHIP siRNA (siCHIP) for 48 h and then treated with TM. CHIP knockdown significantly increased the protein levels of TM-induced UPR-related proteins GRP78, ATF6, and XBP-1s (Figure 1a), and cleaved forms of PARP and caspase-3 were also increased by CHIP knockdown (Figure 1b). The mRNA levels of ATF4, CHOP, and ATF6 were also increased by CHIP knockdown (Figure 1c). These findings were further confirmed in primary hepatocytes isolated from CHIP^+/+^, CHIP^+/−^, and CHIP^−/−^ mice where in the TM-induced ER stress response, the protein levels of UPR-related proteins were enhanced by deficiency of CHIP (Figure 1d). In addition, the ER stress inducer brefeldin A also increased the protein levels of GRP78, CHOP, and cleaved PARP-1 in primary hepatocytes (Figure 1e). These findings suggest that CHIP deficiency accelerates ER stress and apoptosis in hepatocytes.

### 3.2. CHIP Protects Mice from Tunicamycin-Induced Hepatic Steatosis

To determine whether CHIP plays a role in TM-induced hepatic steatosis in vivo, CHIP^+/+^ and CHIP^+/−^ mice were challenged with TM (2 μg/g body weight, i.p.) for 8 h or 36 h, and histological analyses were performed on liver tissue sections from TM-injected CHIP^+/+^ mice and CHIP^+/−^ mice. Lipid accumulation was assessed by Oil Red O staining. Hepatic lipid accumulation was significantly enhanced in the livers of CHIP^+/−^ mice than those of CHIP^+/+^ mice 36 h after the TM challenge (Figure 2a). H&E staining of liver sections also showed central vein obstruction after TM treatment (Figure 2a). Furthermore, the livers of CHIP^+/−^ mice showed enhanced mRNA levels of ATF6, ATF4, and CHOP compared with those of CHIP^+/+^ mice (Figure 2b). Similarly, the protein levels of CHOP and cleaved caspase-3 were higher in CHIP^+/−^ mice than in CHIP^+/+^ mice at 8 h after TM injection; higher levels of GRP94, ACC, and FAS were also found in CHIP^+/−^ mice at 36 h after TM injection (Figure 2c,d). These results suggest that CHIP haploinsufficiency increases TM-induced ER stress responses and aggravates hepatic steatosis.

### 3.3. CHIP Protects Mice from Diet-Induced Hepatic Steatosis

Diets-induced hepatic steatosis is known to be associated with ER stress [39], and thus, we investigated whether CHIP is involved in diet-induced hepatic steatosis. CHIP^+/+^ and CHIP^+/−^ mice were fed chow (10% fat) or the HF (60% fat) diet for 12 weeks. CHIP^+/−^ mice fed the HF diet showed enhanced body weight gains compared to CHIP^+/+^ mice fed the HF diet (Figure 3a), even though the HF diet-exposed mice consumed similar amounts of food as the chow diet-exposed mice (Figure 3b). Significantly enhanced impairment in glucose tolerance was also observed in CHIP^+/−^ mice fed the HF diet compared to CHIP^+/+^ mice (Figure 3c). H&E staining and Oil Red O staining revealed the accumulation of larger lipid droplets in the livers of CHIP^+/−^ mice fed the HF diet compared with CHIP^+/+^ mice (Figure 3d). It has been reported that insulin-stimulated Akt phosphorylation is highly correlated with systemic insulin sensitivity [40]. The levels of Akt phosphorylation in liver tissues of CHIP^+/+^ and CHIP^+/−^ mice fed the HF were measured 10 min after insulin injection. The phosphorylation level of Akt was found to be significantly lower in CHIP^+/−^ mice (Figure 3e).

Additionally, to induce NASH, CHIP^+/+^ and CHIP^+/−^ mice were fed chow (11.5% fat, 0% sucrose) or HF–HS (36% fat, 30% sucrose) diet for 22 weeks, and body and fat weight, IPGTT, and IPITT were measured. CHIP^+/−^ mice fed with HF–HS diet showed enhanced body weight gain and fat weight than CHIP^+/+^ mice (Figure 4a,c). In the high-fat diet group, CHIP^+/−^ mice had higher food intake than CHIP^+/+^ mice (Figure 4b). To determine whether CHIP is involved in HF–HS-induced glucose abnormalities and insulin resistance, IPGTT and IPITT were performed. CHIP^+/−^ mice on HF–HS diet showed significantly impaired glucose tolerance and insulin resistance compared to CHIP^+/+^ mice (Figure 4d,e). Histological analysis of H&E-stained liver sections revealed the pathological changes of hepatic steatosis in the livers of CHIP^+/−^ mice fed HF–HS diet (Figure 4f, top), and Oil Red O staining showed significantly more lipid accumulation in the livers of CHIP^+/−^ mice fed HF–HS diet than CHIP^+/+^ mice fed HF–HS diet (Figure 4f, bottom). Similarly, enhanced protein levels of lipogenic markers (ACC and FAS) and reduced levels of a fatty acid oxidation marker (PGC1α) were observed in CHIP^+/−^ mice fed HF–HS diet compared with CHIP^+/+^ mice fed HF–HS diet (Figure 4g). Furthermore, it showed that the transcriptional alteration of mRNA levels of genes involved in lipogenesis (ACC), fatty acid oxidation (PPARα) and lipid uptake (CD36, FATP1) by the HF–HS diet was significantly reversed in liver tissues from CHIP^+/−^ mice, but not APOB, a marker of the VLDL secretion pathway (Figure 4h). Collectively, these results suggest that CHIP protects mice against diet-induced hepatic steatosis and insulin resistance via inhibiting metabolic dysregulation.

### 3.4. CHIP Protects against Diet-Induced Oxidative Stress and Inflammasome Formation

It has been reported that TXNIP is closely related to oxidative stress and inflammasome activation [41]. MDA is a product of lipid peroxidation and is a representative oxidative stress marker. TXNIP is directly bind to the Nod-like receptor protein 3 (NLRP3) inflammasome in a ROS-dependent manner after its detachment from thioredoxin [41]. In addition, it has been reported that oxidative stress and inflammasomes are key points in the progression from NAFLD to NASH [42]. Therefore, to investigate whether CHIP plays a role in oxidative stress and inflammasome formation, MDA concentrations and NRLP3 protein levels were confirmed in HF–HS diet-induced livers. The hepatic MDA concentrations and NLRP3 protein levels increased by the HF–HS diet were significantly elevated in CHIP^+/−^ mice livers compared to CHIP^+/+^ livers (Figure 5a,b). These data suggest that CHIP inhibits oxidative stress and inflammasome activation, which is associated with TXNIP.

### 3.5. CHIP Promotes the Ubiquitin-Dependent Degradation of TXNIP

Since TXNIP is a critical regulator of ER stress, glucose homeostasis, cholesterol accumulation, and fatty acid synthesis in the liver and CHIP is a E3 ubiquitin ligase [19], we assessed the direct relationship between CHIP and TXNIP via ubiquitin-dependent protein degradation. To determine whether CHIP regulates TXNIP expression through posttranslational modifications, the effects of CHIP on TXNIP expression were measured in HepG2 cells. The protein levels of TXNIP were decreased by CHIP in a dose-dependent manner (Figure 6a), and the immunoprecipitation analysis further confirmed that TXNIP interacts with CHIP (Figure 6b). To determine whether the ubiquitin/proteasome system mediates CHIP-induced TXNIP degradation, the effect of the proteasome inhibitor MG132 on CHIP-mediated TXNIP degradation was measured. TXNIP expression was markedly decreased by CHIP, and MG132 abrogated the CHIP-induced TXNIP degradation (Figure 6c). Because protein degradation by proteasomes is largely dependent on ubiquitination of target substrates, we investigated whether TXNIP is ubiquitinated by CHIP. Overexpression of CHIP markedly increased ubiquitination of TXNIP (Figure 6d), and TXNIP ubiquitination was greatly diminished by E3 ligase activity-defective mutant CHIP H260Q (Figure 6e), which suggest that CHIP-mediated TXNIP degradation is dependent on its Ub ligase activity. Taken together, our data clearly demonstrate that TXNIP expression is regulated by CHIP-mediated ubiquitination.

### 3.6. CHIP Regulates ER Stress and Inflammatory Responses by Inhibiting TXNIP

To determine whether CHIP-mediated TXNIP inhibition is involved in CHIP-dependent regulation of hepatic steatosis, the effects of TXNIP knockdown was evaluated. TM-induced ER stress responses and brefeldin A-induced inflammatory responses were enhanced by CHIP inhibition, and it was alleviated by inhibition of TXNIP using siTXNIP (Figure 7a,b). In addition, TXNIP expression was increased in the livers of TM- and diet-induced NAFLD models (Figure 7c,d). In particular, TXNIP-positive signals were mainly found in hepatic stellate cell-like cells in the livers of diet-induced NAFLD models. These results suggest that CHIP inhibits ER stress, inflammatory responses, and hepatic steatosis by inhibiting hepatic TXNIP expression.

## 4. Discussion

Recent studies have reported the regulation of TXNIP protein at the post-translational level [43]. However, the post-translational regulatory mechanisms of TXNIP are not well understood yet. TXNIP is a member of the α-arrestin protein family that contains two distinctive arrestin-like domains and two PPxY motifs in the C-terminal tail [44]. Zhang et al. demonstrated that the E3 ubiquitin ligase Itch mediates polyubiquitination of TXNIP. TXNIP and Itch interact via the WW domain and the PPXY motif [45]. CHIP contains three tetratricopeptide repeats at the N terminus, a middle dimerization domain, and a U-box at the C terminus [34]; the U-box domain has intrinsic ubiquitin E3 ligase activity, which promotes the ubiquitination of CHIP-bound target proteins [34,46]. CHIP H260Q mutants carry a point mutation in the U-box domain that interferes with E3 ubiquitin ligase activity [47]. In the present study, we found that CHIP promotes the ubiquitin-dependent degradation of TXNIP (Figure 6), which provides new insight into the molecular mechanisms underlying the regulation of ER stress and diet-induced hepatic steatosis and demonstrates the role of CHIP as a novel therapeutic gene for treating hepatic steatosis. The identification of a CHIP region that regulates TXNIP expression may reveal promising therapeutic strategies for the treatment of hepatic steatosis. Therefore, future studies will focus on investigating the binding regions of CHIP responsible for its interactions with TXNIP.

Previous reports have shown that ER stress exacerbates hepatic lipidosis [39], which has also been confirmed in the present study. In addition, the present study showed that steatosis is exacerbated in CHIP^+/−^ mice, and provided an underlying molecular mechanism whereby CHIP regulates ER stress-mediated NAFLD in mice.

Recent studies have shown that ER stress and UPR signaling are associated with hepatic steatosis, which is due to either increased lipogenesis or decreased hepatic lipoprotein secretion [48]. Lee et al. reported that hepatic IRE1α/XBP1 controls the expression of lipogenic enzymes (SCD1, ACC2, and DGAT2), which are crucial for fatty acid and cholesterol biosynthesis [49], whereas the IRE1α and/or ATF6 play a role in preventing ER stress-dependent hepatic steatosis [50,51]. Moreover, PERK/eIF2a is required for the expression of lipogenic genes and progression of hepatic steatosis [52]. However, the underlying mechanisms linking ER stress to hepatic steatosis are not fully understood yet. Previous studies have reported that the kidneys of male mice are typically significantly more sensitive to ER stress-induced kidney damage than those of females [17]. Thus, in this study, female mice were used to focus on TM-induced hepatic steatosis. The present study showed that haploinsufficiency of CHIP accelerates TM-induced ER stress and hepatic steatosis in vivo (Figure 2).

The current prevalence of obesity and related metabolic disorders are closely associated with excessive consumption of HF–HS foods called the western-style diet. The classic western diet is high in both saturated fat and sugar, and has been related to the development of NAFLD. Previous studies demonstrated that the effects of HF and/or HS diets on metabolic risk factors [53,54,55]. A recent report showed that the rapid onset of hepatic steatosis, adipose tissue hypertrophy, and hyperinsulinemia by ingestion of a HF–HS diet may be due to the rapid response of insulin signaling, lipogenesis, and inflammatory genes [56]. In addition, it has also been reported that high-fat diets could only induce steatosis and that the HF–HS diet generated severe steatosis with inflammation, oxidative stress, and myofibroblast and collagen deposition associated with increased serum AST and ALT levels [57]. In this study, haploinsufficiency of CHIP accelerated HF–HS diet-induced weight gain, increased fat mass, impairment of glucose tolerance and insulin resistance, and hepatic steatosis (Figure 4).

Hepatic triglyceride accumulation is a hallmark of NAFLD and results from an imbalance in lipid content between lipid acquisition (de novo lipogenesis, fatty acid absorption) and removal (fatty acid oxidation, VLDL secretion) [58]. It is not clear whether CHIP regulates the progression of NAFLD through the lipid metabolic pathway. Hepatic stellate cells (HSCs) play a critical role in fibrogenesis, and are known to contribute to pathways of inflammation and tissue injury, especially in NASH [59]. In addition, it was reported that inhibition of TXNIP expression can suppress HSC activation in the LX-2 cell line [60]. It was confirmed that the expression of TXNIP was increased in HSCs of CHIP-insufficient mouse liver tissues (Figure 7d). This suggests that CHIP could suppress hepatic fibrosis by regulating TXNIP expression in HSCs.

Further study is required to understand the underlying molecular mechanisms through which TXNIP is regulated by CHIP and how CHIP tightly controls the suppressive response to cirrhosis. Taken together, these findings indicate that CHIP protects against TM- or diet-induced NAFLD and is a potential therapeutic target for the treatment of liver diseases.

## Figures and Tables

**Figure 1 antioxidants-12-00458-f001:**
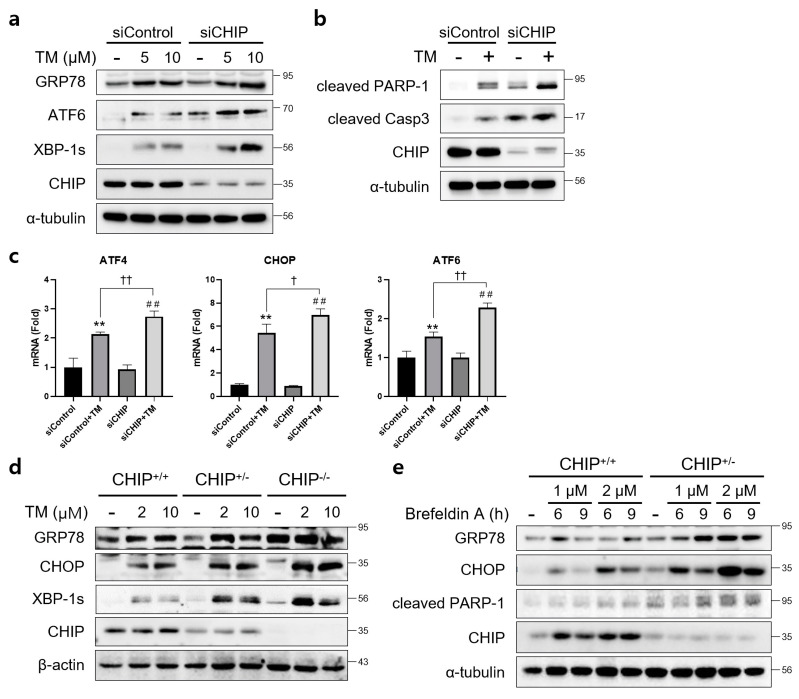
CHIP is involved in unfolded protein responses (UPRs) and apoptosis in hepatocytes. (**a**) AML12 cells were transfected with control (siControl) or CHIP siRNA (siCHIP) for 48 h and then treated with tunicamycin (TM, 5 or 10 μM) for 24 h. Protein levels of GRP78, ATF6, XBP-1s and CHIP were determined by immunoblotting. A-tubulin was used as a loading control. (**b**) AML12 cells were transfected with siControl or siCHIP for 48 h and then treated with TM (10 μM) for 24 h. Protein levels of cleaved PARP-1, cleaved caspase-3 (cleaved Casp3), and CHIP were determined by immunoblotting. α-tubulin was used as a loading control. (**c**) AML12 cells were transfected with siControl or siCHIP for 48 h and then treated with TM (10 μM) for 24 h. Expression of UPR-related genes were measured by qRT-PCR. Relative expression levels were normalized to GAPDH levels. ** *p* < 0.01 vs. siControl, ^##^ *p* < 0.01 vs. siCHIP, ^†^ *p* < 0.05 and ^††^ *p* < 0.01. (**d**) Primary hepatocytes from CHIP^+/+^, CHIP^+/−^, and CHIP^−/−^ mice were treated with TM (2 or 10 μM) for 6 h. Protein levels of GRP78, CHOP, XBP-1s, and CHIP were measured by immunoblotting. β-actin was used as a loading control. (**e**) Primary hepatocytes from CHIP^+/+^ and CHIP^+/−^ mice were treated with brefeldin A (BFA, 1 or 2 μM) for 6 or 9 h. Protein levels of GRP78, CHOP, cleaved PARP-1, and CHIP were measured by immunoblotting. α-tubulin was used as a loading control.

**Figure 2 antioxidants-12-00458-f002:**
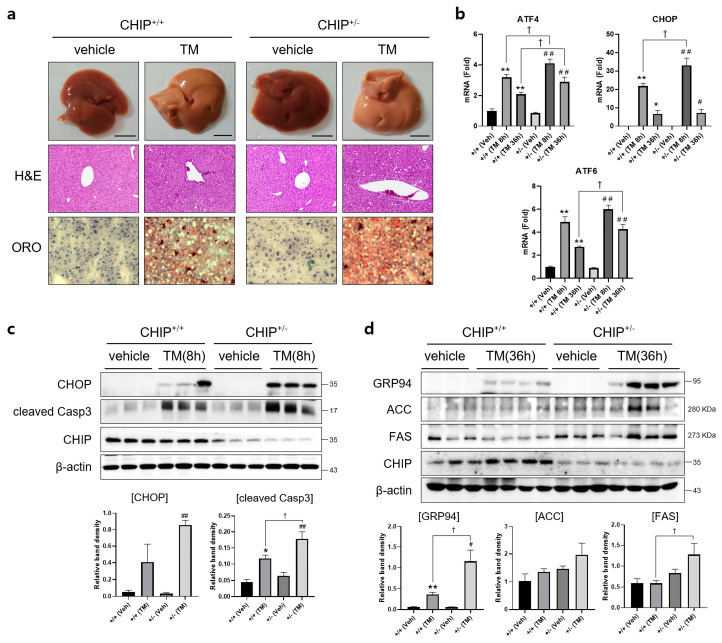
CHIP is responsible for tunicamycin-induced hepatic steatosis via UPR and apoptosis in mice. CHIP^+/+^ and CHIP^+/−^ mice were injected intraperitoneally with tunicamycin (TM, 2 μg/g body weight) and sacrificed 8 h or 36 h (n = 5) after TM injection. (**a**) Representative picture of whole mouse livers (top). Liver tissue sections were stained with H&E (middle) and Oil Red O (bottom) (original magnification, ×200). (**b**) Expression levels of UPR-related genes in livers of the CHIP^+/+^ and CHIP^+/−^ mice were measured by qRT-PCR. Relative expression levels were normalized to GAPDH levels. * *p* < 0.05 and ** *p* < 0.01 vs. CHIP^+/+^ mice (vehicle), ^#^ *p* < 0.05 and ^##^ *p* < 0.01 vs. CHIP^+/−^ mice (vehicle). ^†^ *p* < 0.05. (**c**,**d**) Protein levels of CHOP, cleaved Casp3, GRP94, ACC, FAS, and CHIP in liver tissues were measured by immunoblotting. β-actin was used as a loading control.

**Figure 3 antioxidants-12-00458-f003:**
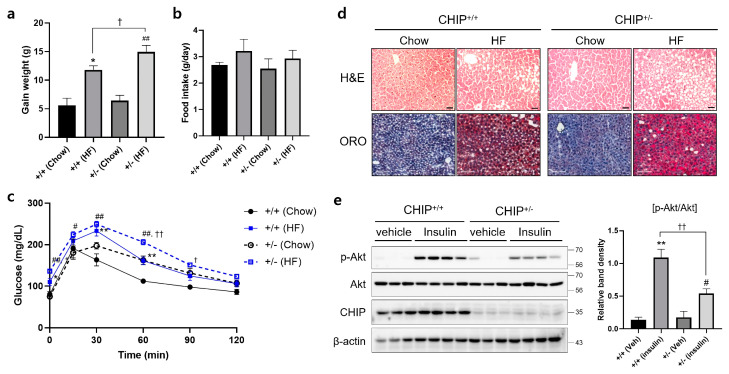
CHIP protects from high-fat diet-induced hepatic steatosis and insulin resistance in mice. CHIP^+/+^ and CHIP^+/−^ mice were fed chow (10% fat) or the high-fat (HF, 60% fat) diet for 12 weeks (n = 5–7). (**a**) Average weight gain after 12-week feeding schedule. * *p* < 0.05 vs. CHIP^+/+^ mice with Chow, ^##^ *p* < 0.01 vs. CHIP^+/−^ mice with Chow. ^†^ *p* < 0.05. (**b**) Average daily food intake of CHIP^+/+^ and CHIP^+/−^ mice during experiment. (**c**) Glucose tolerance test was conducted by intraperitoneal injection of glucose (2 g/kg body weight). Blood glucose was measured at 0, 15, 30, 60, 90, and 120 min after glucose injection. * *p* < 0.05 and ** *p* < 0.01 vs. CHIP^+/+^ mice with Chow, ^#^ *p* < 0.05 and ^##^ *p* < 0.01 vs. CHIP^+/−^ mice with Chow. ^†^ *p* < 0.05 and ^††^ *p* < 0.01 vs. CHIP^+/+^ mice with HF. (**d**) Liver tissue sections were stained with H&E (top) and Oil Red O (bottom) (original magnification, ×200). (**e**) HF diet-fed CHIP^+/+^ and CHIP^+/−^ mice were injected intraperitoneally with insulin (1.5 U/kg body weight) and sacrificed 10 min after insulin injection. Liver tissues were harvested immediately, proteins were extracted, and immunoblotting was conducted to measure expression levels of p-Akt, Akt, and CHIP. β-actin was used as a loading control. Relative p-Akt levels were normalized versus Akt. ** *p* < 0.01 vs. CHIP^+/+^ mice with vehicle (Veh), ^#^ *p* < 0.05 vs. CHIP^+/−^ with Veh, ^††^ *p* < 0.01.

**Figure 4 antioxidants-12-00458-f004:**
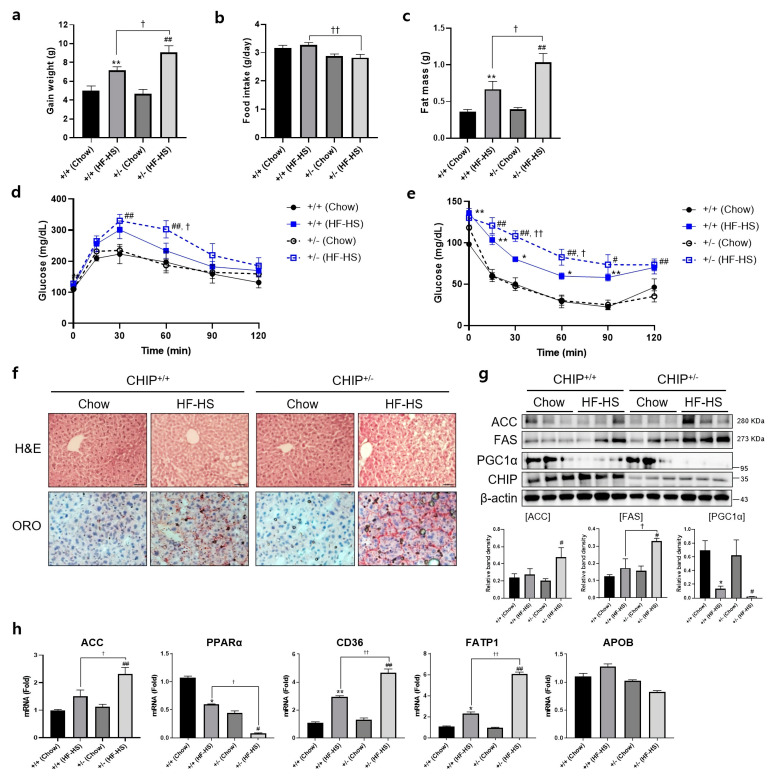
CHIP protects from high-fat and high-sucrose diet-induced hepatic steatosis in mice. CHIP^+/+^ and CHIP^+/−^ mice were fed chow (11.5% fat, 0% sucrose) or the high-fat–high-sucrose (HF–HS, 36% fat, 30% sucrose) diet for 22 weeks (n = 5–7). (**a**) Average weight gain after 22-week feeding schedule. ** *p* < 0.01 vs. CHIP^+/+^ mice with Chow, ^##^ *p* < 0.01 vs. CHIP^+/−^ mice with Chow. ^†^ *p* < 0.05. (**b**) Average daily food intake of CHIP^+/+^ and CHIP^+/−^ mice during experiment. ^††^ *p* < 0.01. (**c**) Gonadal fat pad was harvested and weighed as fat mass. ** *p* < 0.01 vs. CHIP^+/+^ mice with Chow, ^##^ *p* < 0.01 vs. CHIP^+/−^ mice with Chow. ^†^ *p* < 0.05. (**d**) Glucose tolerance tests were conducted by intraperitoneal injection of glucose (2 g/kg body weight). Blood glucose levels were measured at 0, 15, 30, 60, 90, and 120 min after glucose injection. ^##^ *p* < 0.01 vs. CHIP^+/−^ mice with Chow. ^†^ *p* < 0.05 vs. CHIP^+/+^ mice with HF–HS. (**e**) Insulin tolerance test was conducted by intraperitoneal injection of insulin (0.3 U/kg body weight). Blood glucose was measured at 0, 15, 30, 60, 90, and 120 min after insulin injection. * *p* < 0.05 and ** *p* < 0.01 vs. CHIP^+/+^ mice with Chow, ^#^ *p* < 0.05 and ^##^ *p* <0.01 vs. CHIP^+/−^ mice with Chow. ^†^ *p* < 0.05 and ^††^ *p* < 0.01 vs. CHIP^+/+^ mice with HF–HS. (**f**) Liver tissue sections were stained with H&E (top) and Oil Red O (bottom) (original magnification, ×200). (**g**) Protein levels of ACC, FAS, PGC1α, and CHIP in liver tissues were measured by immunoblotting. β-actin was used as a loading control. * *p* < 0.05 vs. CHIP^+/+^ mice with Chow, ^#^ *p* < 0.05 vs. CHIP^+/−^ with Chow, ^†^ *p* < 0.05. (**h**) mRNA expression of genes involved in lipogenesis (ACC), fatty acid oxidation (PPARα), lipid uptake (CD36, FATP1), and VLDL secretion (APOB) were measured by qRT-PCR. Relative expression levels were normalized to GAPDH levels. * *p* < 0.05 and ** *p* < 0.01 vs. CHIP^+/+^ mice with Chow, ^#^ *p* < 0.05 and ^##^ *p* <0.01 vs. CHIP^+/−^ mice with Chow. ^†^ *p* < 0.05 and ^††^ *p* < 0.01 vs. CHIP^+/+^ mice with HF–HS.

**Figure 5 antioxidants-12-00458-f005:**
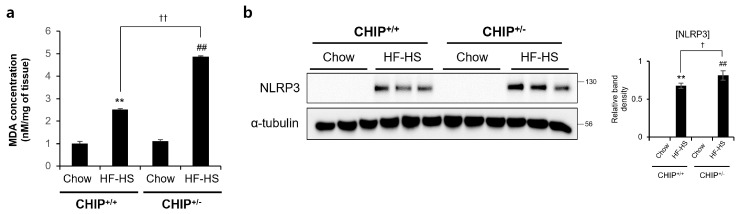
CHIP is responsible for high-fat and high-sucrose diet-induced oxidative stress and inflammasome formation. CHIP^+/+^ and CHIP^+/−^ mice were fed chow (11.5% fat, 0% sucrose) or the high-fat–high-sucrose (HF–HS, 36% fat, 30% sucrose) diet for 22 weeks (n = 5–7). (**a**) MDA concentrations in liver tissues were measured by ELISA method. ** *p* < 0.01 vs. CHIP^+/+^ mice with Chow, ^##^ *p* < 0.01 vs. CHIP^+/−^ mice with Chow. ^††^ *p* < 0.01. (**b**) Protein levels of NLRP3 in liver tissues were measured by immunoblotting. α-tubulin was used as a loading control. ** *p* < 0.01 vs. CHIP^+/+^ mice with Chow, ^##^ *p* < 0.01 vs. CHIP^+/−^ mice with Chow. ^†^ *p* < 0.05.

**Figure 6 antioxidants-12-00458-f006:**
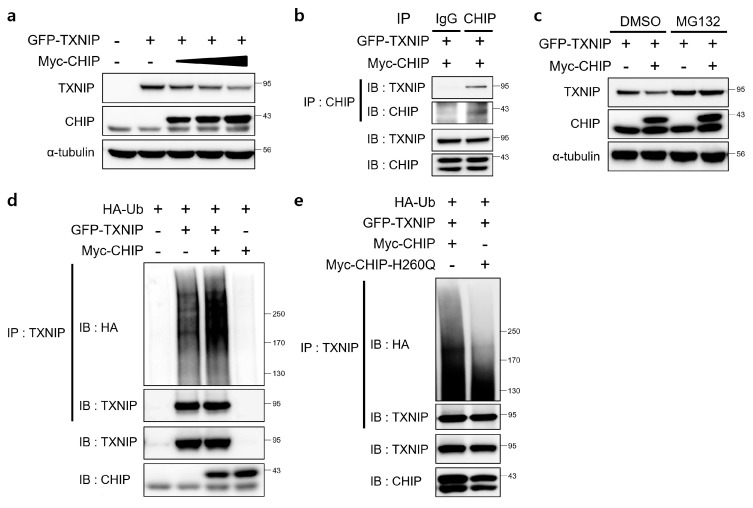
E3 ligase CHIP promotes ubiquitination of TXNIP. (**a**) HepG2 cells were transfected with GFP-tagged TXNIP (GFP-TXNIP) with or without Myc-tagged CHIP (Myc-CHIP) for 24 h. Protein levels of TXNIP and CHIP were measured by immunoblotting. α-tubulin was used as a loading control. (**b**) HepG2 cells were transfected with GFP-TXNIP and MYC-CHIP for 24 h. Cells were treated with MG132 (10 μM) for 6 h prior to lysis and immunoprecipitated with anti-CHIP antibody. Protein levels of TXNIP and CHIP were measured by immunoblotting. (**c**) HepG2 cells were transfected with GFP-TXNIP with or without Myc-CHIP for 24 h. Cells were treated with MG132 (10 μM) for 6 h prior to lysis, and protein levels of TXNIP and CHIP were measured by immunoblotting. α-tubulin was used as a loading control. (**d**,**e**) HepG2 cells were co-transfected with HA-tagged ubiquitin (HA-Ub), GFP-TXNIP and Myc-CHIP or Myc-CHIP-H260Q for 24 h. Cells were treated with MG132 (10 μM) for 6 h prior to lysis and immunoprecipitated with anti-TXNIP antibody. Amounts of polyubiquitinated proteins in cell lysates were measured by immunoblotting with anti-HA antibody. Protein levels of TXNIP and CHIP were measured.

**Figure 7 antioxidants-12-00458-f007:**
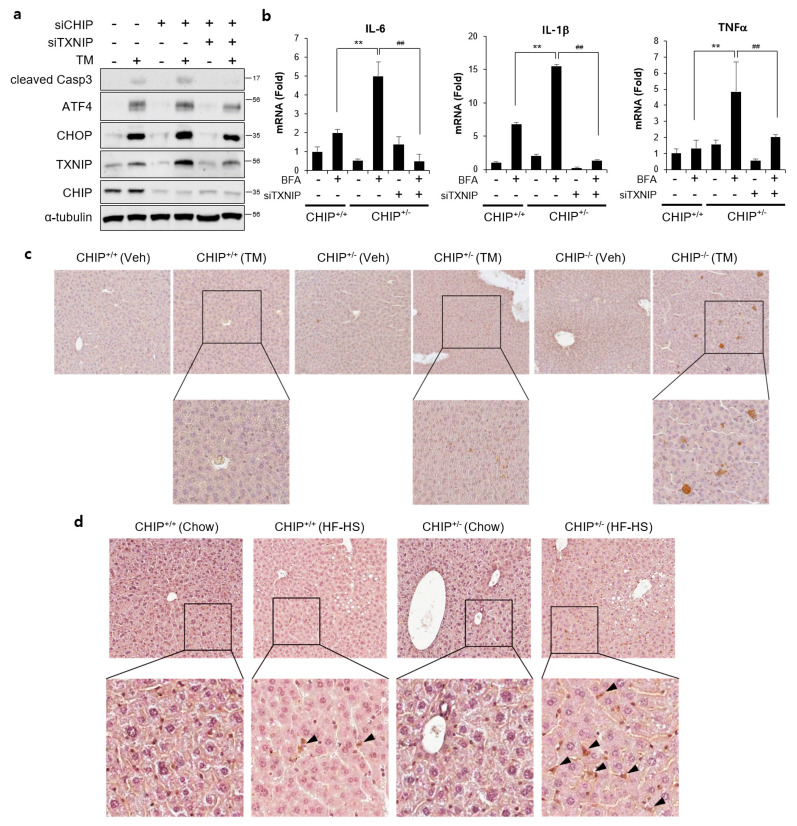
CHIP-mediated TXNIP regulation is responsible for ER stress and inflammatory response. (**a**) HepG2 cells were transfected with CHIP siRNA (siCHIP) and/or TXNIP siRNA (siTXNIP) for 48 h and then treated with tunicamycin (TM, 5 μM) for 8 h. Protein levels of cleaved Casp3, ATF4, CHOP, TXNIP, and CHIP were measured by immunoblotting. α-tubulin was used as a loading control. (**b**) Primary hepatocytes from CHIP^+/+^ and CHIP^+/−^ mice were transfected with siTXNIP for 48 h and then treated with brefeldin A (BFA, 1 μM) for 6 h. mRNA expression of inflammatory genes (IL-6, IL-1β, and TNFα) were measured by qRT-PCR. Relative expression levels were normalized to GAPDH levels. ** *p* < 0.01 vs. siControl, ^##^ *p* < 0.01 vs. siCHIP. (**c**) CHIP^+/+^, CHIP^+/−^, and CHIP^−/–^ mice were injected intraperitoneally with TM (2 μg/g body weight) and sacrificed 36 h after TM injection. TXNIP expression in liver tissue sections was determined by immunohistochemistry (IHC). (**d**) CHIP^+/+^ and CHIP^+/−^ mice were fed chow or HF–HS diet for 22 weeks. Representative images of liver sections from CHIP^+/+^ and CHIP^+/−^ mice following HF–HS diet fed. TXNIP expression in liver tissue sections was determined by IHC.

## Data Availability

The data are contained within the article.

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
