# Peer review of "CHIP Haploinsufficiency Exacerbates Hepatic Steatosis via Enhanced TXNIP Expression and Endoplasmic Reticulum Stress Responses"

_antioxidants, 2023, doi:10.3390/antiox12020458_

Round 1
Reviewer 1 Report
In the article ‘CHIP haploinsufficiency exacerbates experimental hepatic steatosis via enhanced TXNIP expression and ER stress responses’, there are some critical questions.
1. The phenotypes of steatosis of mice are very confused. When reading through this article, the HE staining results and Oil red O staining are inconsistent in the whole article, especially the HE staining showing no lipid droplets in the liver. For example, the H&E staining images of Figure 2a, 3e and 4f.
2. All the figures have poor quality, especially the images of the western blotting of Figure 1a, 1d, 1e, 2c and 4g. It should be improved or redo.
3. The bands of the western blotting should label molecular weight.
4. In the fields of steatosis, lipid beta-oxidation pathway, uptake pathway and VLDL pathway are very important and should be considered besides lipogenesis pathway. Western blotting and q-PCR analysis need to be performed.
Author Response
Response to Reviewer 1 Comments
In the article ‘CHIP haploinsufficiency exacerbates experimental hepatic steatosis via enhanced TXNIP expression and ER stress responses’, there are some critical questions.
1. The phenotypes of steatosis of mice are very confused. When reading through this article, the H&E staining results and Oil red O staining are inconsistent in the whole article, especially the HE staining showing no lipid droplets in the liver. For example, the H&E staining images of Figure 2a, 3e and 4f.
Re: We agree reviewer’s comment and carefully evaluated steatosis with tissue images. Although a mouse model of tunicamycin-induced steatosis is relatively weak compared to high fat diet-induced steatosis, tunicamycin-induced accumulation of lipid droplets was markedly enhanced in liver tissues from CHIP+/- compared with wild type littermate in new Figure 2a. In addition, we replaced H&E images with high magnification to reveal better resolution in the revised manuscript (Fig. 3d and 4f). These results suggest that CHIP negatively regulate hepatic steatosis.
2. All the figures have poor quality, especially the images of the western blotting of Figure 1a, 1d, 1e, 2c and 4g. It should be improved or redo.
Re: As reviewer’s suggestion, we improved figure resolution quality and replaced the Western blot images by using original samples in the revised manuscript (Fig. 1a, Fig. 1d, Fig. 2c, Fig. 4g).
3. The bands of the western blotting should label molecular weight.
Re: As reviewer’s suggestion, we did label molecular weight of Western blot images in the revised manuscript.
- In the fields of steatosis, lipid beta-oxidation pathway, uptake pathway and VLDL pathway are very important and should be considered besides lipogenesis pathway. Western blotting and q-PCR analysis need to be performed.
Re: We agree reviewer’s comment and determined expressions of various molecules involved in lipid metabolism. As shown in new Figure 4g and 4h, diet-induced molecular alterations of lipogenesis, lipid beta-oxidation, and lipid uptake pathway were enhanced in CHIP+/- compared with wild type littermate, but not VLDL pathway. These results suggest that CHIP protects mice against diet-induced hepatic steatosis via inhibiting metabolic dysregulation. We added these results in the revised manuscript (new Fig. 4g & 4h).
Reviewer 2 Report
The manuscript is very interesting. The authors have done robust experimental work. The number of experiments is remarkable. The methodology used is updated and sufficient. The results support the discussion, and the manuscript is well written. I only have a few minor comments.
I. Minor comments:
1. In the title I suggest not using the abbreviation "RE" and eliminate the word experimental
I suggest the suggested title:
CHIP haploinsufficiency exacerbates hepatic steatosis via enhanced TXNIP expression and endoplasmic reticulum stress responses
2. Improve the wording of the objective of the study.
3. In the introduction, I suggest including a brief paragraph on the importance of diet in the development of NAFLD. PMID: 26512643
4. In the introduction, it would be interesting to briefly discuss the effect of mitochondrial dysfunction on the development of hepatic steatosis. PMID: 32620521
Author Response
Response to Reviewer 2 Comments
The manuscript is very interesting. The authors have done robust experimental work. The number of experiments is remarkable. The methodology used is updated and sufficient. The results support the discussion, and the manuscript is well written. I only have a few minor comments.
Minor comments:
1. In the title I suggest not using the abbreviation "ER" and eliminate the word experimental. I suggest the suggested title: “CHIP haploinsufficiency exacerbates hepatic steatosis via enhanced TXNIP expression and endoplasmic reticulum stress responses’
Re: We agree reviewer’s comment and changed title like as reviewer’s suggestion in the revised manuscript.
2. Improve the wording of the objective of the study.
Re: As reviewer’s suggestion, we tried to improve the wording of this study’s objective in the revised manuscript.
3. In the introduction, I suggest including a brief paragraph on the importance of diet in the development of NAFLD. PMID: 26512643
Re: As reviewer’s suggestion, we described the importance of diet in the development of NAFLD in Introduction section in the revised manuscript.
4. In the introduction, it would be interesting to briefly discuss the effect of mitochondrial dysfunction on the development of hepatic steatosis. PMID: 32620521
Re: As reviewer’s suggestion, we described the effect of mitochondrial dysfunction on the development of hepatic steatosis in Introduction section in the revised manuscript.
Reviewer 3 Report
Thank you very much. The authors said CHIP protects against TM or HF-HS diet induced NAFLD and suggest as a potential therapeutic target for the treatment of liver disease. This paper covered many ER stress factors and reactions, and had many experiments and data and analyzed by CHIP +/- mice and +/+ mice. This paper was excellent and will be useful of ER stress research.
Author Response
Response to Reviewer 3 Comments
Thank you very much. The authors said CHIP protects against TM or HF-HS diet induced NAFLD and suggest as a potential therapeutic target for the treatment of liver disease. This paper covered many ER stress factors and reactions, and had many experiments and data and analyzed by CHIP +/- mice and +/+ mice. This paper was excellent and will be useful of ER stress research.
Re: Thank you very much for your encouraging comments which helped us further pursue our research interest in the field of ER stress and metabolic diseases.
Round 2
Reviewer 1 Report
I have no question currently.